# From non-use to covert and overt use of contraception: Identifying community and individual factors informing Nigerian women's degree of contraceptive empowerment

**Funmilola M. OlaOlorun**[1]*, **Philip Anglewicz**[2], **Caroline Moreau**[2,3]

**1** Department of Community Medicine, College of Medicine, University of Ibadan, Ibadan, Nigeria,
**2** Department of Population, Family and Reproductive Health, Johns Hopkins Bloomberg School of Public Health, Baltimore, Maryland, United States of America, **3** Soins et Santé Primaire, CESP Centre for Research in Epidemiology and Population Health U1018, Inserm, Villejuif, France

* fmolaolorun@gmail.com

## Abstract

### Objective

In Nigeria, unmet need for contraception is high despite improved access to modern contraception. To identify factors that support Nigerian women's contraceptive decisions to achieve their reproductive goals, in the presence or absence of their partner's support, we seek to identify individual/couple and community level determinants of a spectrum of contraceptive practices, from non-use to covert and overt use of contraception.

### Methods

Data were drawn from a national probability survey conducted by Performance Monitoring and Accountability 2020 in Nigeria in 2017–2018. A sample of 12,948 women 15–49 years was included, 6433 of whom were in need of contraception at the time of the survey. We conducted bivariate and multivariate analysis to identify individual/couple and community level factors associated with covert use relative to non-use and to overt use of contraception.

### Results

Altogether, 58.0% of women in need of contraception were non-users, 4.5% were covert users and 37.5% used contraception overtly. Covert users were more educated and wealthier than non-users, but less educated and less wealthy than overt users. Covert users were less likely to cohabitate with their partner compared to non-users [AOR = 4.60 (95%CI: 3.06–6.93)] and overt users [AOR = 5.01 (95%CI: 3.24–7.76)] and more likely to reside in urban areas. At the community level, covert users were more likely to live in communities with higher contraceptive prevalence and higher levels of female education relative to non-users. They were also more likely to live in communities with higher female employment [AOR = 1.62, (95%CI: 0.96–2.73)] compared to overt users.

**Data Availability Statement:** The data underlying the results presented in the study are third-party data available from https://www.pmadata.org. The

data are available to anyone after completing a brief request form at https://www.pmadata.org/data/available-datasets or from a data repository, https://pma.ipums.org/pma/.

**Funding:** PMA2020 was supported by a grant (OPP1079004) from the Bill and Melinda Gates Foundation (https://www.gatesfoundation.org/). The funders had no role in study design, data collection and analysis, decision to publish, or preparation of the manuscript.

**Competing interests:** The authors have declared that no competing interests exist.

## Conclusion

By identifying individual and community level factors associated with the spectrum from non-use to covert use and overt use of contraception, this study highlights the importance of integrating individual and community interventions to support women's realization of their reproductive goals.

## Background

Covert use, the use of a contraceptive method without the knowledge of a woman's partner, may represent her attempt to maintain reproductive autonomy in the face of reproductive coercion and potential violence by her partner [1–5]. Beyond autonomy, covert use may also be due to misperceptions of partners' views of contraceptive use and desired family size [6,7]. While couple communication could address the discrepancy, studies suggest that many couples do not discuss their fertility goals. When couples negotiate reproductive outcomes, this is done against the backdrop of normative values, and if they disagree, whether this is real or perceived, there needs to be some form of conflict resolution [6]. As a result of the outcome of couple communication, or the absence of any communication, women may decide on their own to use a family planning method, even if this means using a method covertly [2,5,8–10].

Covert use is said to be accompanied by psychosocial costs [5,11,12]. In some settings, covert use of modern contraception denied women of their partners' emotional support, such as when hormonal contraception disrupted menstrual flow and sexual pleasure [5]. Moreover, a man's suspicion that his partner is using a contraceptive method covertly may be raised due to side effects, such as heavy bleeding, weight gain or loss, and lack of libido, or due to a delay in her becoming pregnant. Such suspicions, and the tension they cause may lead to disclosure by some women, ultimately bringing about improvement in communication between partners [10]. On the other hand, it has been reported that if a male partner discovers on his own that a woman is using a method without his approval, he may feel betrayed [5,10] and could report her to her parents, accuse her of infidelity, stop having sex with her, start an extramarital affair, physically abuse her, marry another wife, withdraw economic support, or even divorce her, all in a bid to "punish" and "disgrace" her for her actions [5,8–10,12,13].

Despite the social cost, covert use is a common practice. Current estimates are derived from either discordant couple responses or direct questioning of women in large population-based surveys. A study across 21 DHS surveys indicates large discrepancies between these two measures, with direct estimates averaging 6% while indirect estimates average 39%. Most of the discrepancy related to male partners reporting a different contraceptive status than their female partners, despite women indicating that their partners knew about their family planning practices, suggesting over estimation of covert use using the indirect measure [14]. Direct estimates of covert use in clinic-based samples of family planning users from Nigeria were 4.9% in Enugu [15], 6.8% in Ibadan [16] and 7.2% in Ilorin, Nigeria [17]. A qualitative study conducted across cultural settings in Ethiopia, Nigeria, and Uganda also suggests that covert use is a common practice among women [10].

Motivations for covert use of contraception vary by setting. In Nigeria, research suggests that a woman may choose to use contraception covertly due to her partner's disapproval of contraceptive use, his pronatalist outlook, failure to discuss with him, or difficulty in communicating with him about contraception. Moreover, a Nigerian woman may opt for covert contraceptive use because her partner does not provide economic support for his household.

Furthermore, Nigerian women have reported that they chose to use a method covertly due to health concerns or because their childcare responsibilities were taking a toll on them [4,10,13,16].

Characteristics associated with covert use vary across settings. In Uganda, women previously in union and those with no formal education are more likely to report covert use [18]. In an analysis of data from monogamous couples in nine sub Saharan African countries, Gasca and Becker reported that higher education was associated with lower odds of covert use [19]. A study conducted in Ghana also indicates that women who are single and women who wish to wait four or more years before their next birth are more likely to use contraception covertly [5]. The same study indicates that Ghanaian women who were Muslims/Traditionalists were more likely than those who were Christians to report covert use [5]. A Kenyan study suggests that women without reproductive autonomy may depend on covert use of contraception [20]. Experience of physical intimate partner violence has also been said to predict covert use in Ugandan women [18].

One couple-level factor that has consistently been associated with covert use is spousal educational gap. In a study of nine SSA countries, monogamous couples with a wider spousal educational gap reported less covert contraceptive use [19], possibly a function of the woman doing whatever her partner wants as a reflection of the power dynamics. Among contraceptive users in Ibadan, Nigeria, covert use is less likely to occur when a woman receives financial support from her partner and when he approves of contraceptive use [16]. This may be a reflection of couple conversations on fertility desires and family planning, and these have been shown to improve overt use of contraception [21]. On the other hand, qualitative research from Ethiopia, Nigeria, and Uganda suggests that the absence of financial support from her male partner may propel a woman to use a family planning (FP) method covertly [10,13]. The woman may perceive this lack of financial support as an indication of an unstable relationship, and her choice to use a contraceptive covertly may be her own way of self-preservation as she thus avoids another birth that could potentially make her even more dependent on her partner [10,13]. Covert use appears to be more prevalent in rural than urban communities [22], possibly a function of their lower levels of formal education and perceived lack of autonomy or the sense of disenfranchisement that their upbringing ascribes to women.

While there is growing interest in estimating the prevalence, reasons for, and consequences of covert use of contraception to inform effective access to family planning, there is little recognition of the specific positioning of covert use with respect to women's autonomy in reproductive decisions. When compared to non-users of contraception who are in need of contraception, covert users demonstrate some level of empowerment towards achieving their reproductive goals, but at the same time, these women are less likely to voice their preferences compared to overt users of contraception [10]. That said, some researchers have pointed out that covert use is a threat to continued use of contraception and may thus be a risk factor for unintended pregnancy [23,24]. However, research from some sub-Saharan African settings where conversations about sex are taboo, and women are expected to take responsibility for family planning use suggest that in these settings, the choice to use a contraceptive method covertly is evidence of a woman being able to display some form of reproductive autonomy [10].

In this study, we seek to estimate the prevalence of covert use of family planning, as well as individual/couple and community level factors that influence covert use among Nigerian women. In doing so, we identify factors that support women's decision to act on their reproductive goals when they do not wish to become pregnant but hinder their ability to share their decisions with their partners. This question is salient in a context where decades of widespread demand generation efforts and improved access to and supply of modern contraceptive

methods, have resulted in little change in contraceptive prevalence, which remains relatively low while unmet need is high and a significant proportion of women indicate using contraception without the knowledge of their partners [14].

## Methods

Data come from the Performance Monitoring and Accountability 2020 (PMA2020) study, an 11-country mobile phone-assisted survey that collects data on family planning and other reproductive health indicators, as well as information on water, sanitation and hygiene from households, individual women of reproductive age and health service delivery points. Details of the methods of the survey and a description of the sample are available in Zimmerman et al., 2017 [25].

In this research, we conduct a secondary analysis of the most recent surveys by PMA2020—November/December 2017 in Oyo state, and April/May 2018 in seven additional Nigerian states (Anambra, Kaduna, Kano, Lagos, Nasarawa, Rivers, Taraba). A three-stage sampling design was used to select states within geopolitical zones, geographic clusters within each state, and households within geographic clusters. Specifically, within each selected state, enumeration areas (EA) corresponding to geographic clusters containing approximately 200 households were listed, and 35 to 40 households per EA were subsequently randomly selected. All women ages 15 to 49 from the selected households were invited to participate, producing a nationally representative sample of women of reproductive age in Nigeria. In all, 10,070 households (Response rate: 97.5%) and 11,106 *de facto* females (Response rate: 98.1%) provided verbal consent to participate and completed the survey in the 2018 survey. For the 2017 Oyo state survey, there were 2,590 households (Response rate: 97.9%), and 1,842 de facto females (Response rate: 97.0%). The Johns Hopkins Bloomberg School of Public Health and the National Health Research Ethics Committee (NHREC) of Nigeria provided ethical approval for this study.

Data were collected by "Resident Enumerators," trained young women who were non-health workers and resided within or near the study enumeration areas that were selected as study sites. Information obtained from respondents was immediately entered into preprogrammed mobile phones using Open Data Kit software (JHU Collect) and sent to a central server where data were aggregated and anonymized. Interviews were conducted in English, Hausa, Igbo, Pidgin English, or Yoruba, depending on the woman's preference.

For this analysis, we selected women who have a need for contraception and divided them into three groups: non-users of contraception, covert users and overt users of contraception. Women were considered to have a need for contraception if they were currently using or had recently used contraception or if they had not recently used contraception but were sexually active in the last 3 months, were not pregnant and were not trying to become pregnant. Women who desired a child within the 2-year period following the survey were not considered to be in need for contraception and were excluded from the analytic sample. Based on the question, "Does your husband/partner know that you are using [CURRENT METHOD]?", recent or current contraceptive users were either categorized as covert users or overt users of contraception.

Community-level variables were constructed from the available data by averaging individual level data for all women in each enumeration area/community (primary sampling unit), excluding the individual woman's information, and then creating tertiles labeled "lowest", "middle" and "highest" to represent relative measures of the following variables across enumeration areas/communities: (1) women reporting current or recent FP use; (2) women who worked outside the home in the one month preceding the survey; (3) women with secondary level education or higher; (4) women reporting exposure to FP messages through at least one media channel (radio, television, newspapers, billboards/posters, magazines, brochure/leaflet/

flyer, voice or text message). Individual/couple level independent variables included categorical variables: (1) age group (15–24, 25–34, 35–39 years); (2) highest level of education attained (no formal/primary, secondary, tertiary); (3) number of live births (0–1, 2–4, >4); (4) religious affiliation (Catholic, Other Christian, Muslim/Other); (5) marriage type (not living with partner, monogamous, polygynous); and (6) household wealth tertile (lowest, middle, highest); (7) designation of residence as urban, as a proxy for access to health services.

We conducted our analysis in two steps. First, we examined factors related to use of contraception among women in need of contraception who potentially faced barriers to family planning, by comparing covert users to non-users of contraception. In the second phase of the analysis, we examined factors related to covert use compared to overt users of contraception. In each case, couple/individual and community-level factors associated with covert use were assessed using mixed multilevel logistic regression models. Mixed multi-level models allow one to account for clustering of individuals and couples within communities. Random intercepts were allowed, assuming that covert use differs across communities. Data were analyzed using Stata 15.

## Results

Altogether, 3733 (58.0%) of women in need of family planning were non-users, 290 (4.5%) used a method covertly and 2410 (37.5%) used contraception overtly. Covert users were more likely to choose oral pills (24.7%), injectables (21.5%) and implants (19.1%), while overt users favored male condoms (20.8%), injectables (19.9%) and implants (16.3%). As expected, covert users did not report use of male condoms, withdrawal or female sterilization, while overt users did (Fig 1).

Comparing covert users to non-users in need of contraception from bivariate analysis (Table 1), the former were more likely to be older (35–49 years: 43.8% vs 36.7%; p = 0.05) and more educated (tertiary education: 18.3% vs 10.0%; p<0.001). Covert users were more likely than their non-using counterparts not to be living with their partner (38.6% vs 14.3%; p<0.001). Additionally, covert users compared with non-users were richer (middle tertile: 51.4% vs 32.3%; highest tertile: 30.0% vs 22.8%; p<0.001), and more likely to be Catholic or Protestant Christians (Catholic: 15.2% vs 7.3%; Protestant: 41.0% vs 25.0%; p<0.001).

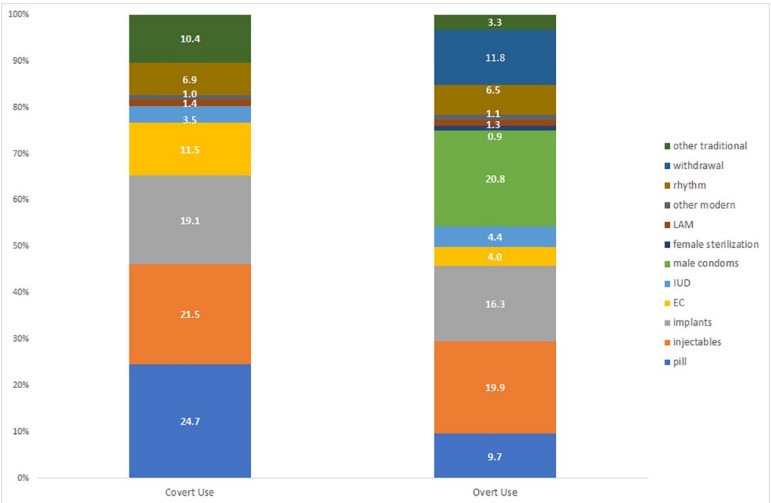

**Fig 1. Method mix for covert and overt users of family planning methods among Nigerian women.** "Other modern" refers to cycle beads, diaphragm and female condom.

**Table 1. Individual-/Couple-level characteristics according to non-use, covert, and overt use of family planning in Nigeria.**

| | Non-users in need of contraception n = 3733 | | Covert Use n = 290 | | Overt Use n = 2410 | | Total N = 6,433 | |
|---|---|---|---|---|---|---|---|---|
| | n | % | n | % | n | % | N | % |
| **Individual-Level Variables** | | | | | | | | |
| **Age group** | | | | | | | | |
| 15-24y | **1052** | **28.2** | **74** | **25.5** | **467** | **19.4** | 1593 | 24.8 |
| 25-34y | **1313** | **35.2** | **89** | **30.7** | **917** | **38.1** | 2319 | 36.1 |
| 35-49y | **1368** | **36.7** | **127** | **43.8** | **1026** | **42.6** | 2521 | 39.2 |
| **Educational attainment** | | | | | | | | |
| No formal/primary | **2123** | **56.9** | **106** | **36.6** | **544** | **22.6** | 2773 | 43.1 |
| Secondary | **1238** | **33.2** | **131** | **45.2** | **1256** | **52.1** | 2625 | 40.8 |
| Tertiary | **372** | **10.0** | **53** | **18.3** | **610** | **25.3** | 1035 | 16.1 |
| **Number of births** | | | | | | | | |
| 0–1 births | 971 | 26.0 | 93 | 32.1 | **606** | **25.2** | 1670 | 26.0 |
| 2–4 births | 1587 | 42.5 | 112 | 38.6 | **1181** | **49.0** | 2880 | 44.8 |
| 4+ births | 1175 | 31.5 | 85 | 29.3 | **623** | **25.9** | 1883 | 29.3 |
| **Religion** | | | | | | | | |
| Catholic | **271** | **7.3** | **44** | **15.2** | 398 | 16.5 | 713 | 11.1 |
| Protestant Christian | **933** | **25.0** | **119** | **41.0** | 1181 | 49.0 | 2233 | 34.7 |
| Islam/Others | **2529** | **67.8** | **127** | **43.8** | 831 | 34.5 | 3487 | 54.2 |
| **Couple-Level Variables** | | | | | | | | |
| **Family type** | | | | | | | | |
| Never married/Separated/Divorced | **534** | **14.3** | **112** | **38.6** | **480** | **19.9** | 1126 | 17.5 |
| Monogamous | **1938** | **51.9** | **119** | **41.0** | **1519** | **63.0** | 3576 | 55.6 |
| Polygynous | **1261** | **33.8** | **59** | **20.3** | **411** | **17.1** | 1731 | 26.9 |
| **Wealth tertile** | | | | | | | | |
| Poorest | **1678** | **45.0** | **54** | **18.6** | **419** | **17.4** | 2151 | 33.4 |
| Middle | **1205** | **32.3** | **149** | **51.4** | **788** | **32.7** | 2142 | 33.3 |
| Richest | **850** | **22.8** | **87** | **30.0** | **1203** | **49.9** | 2140 | 33.3 |
| **Residence** | | | | | | | | |
| Rural | **2251** | **60.3** | **109** | **37.6** | 909 | 37.7 | 3269 | 50.8 |
| Urban | **1482** | **39.7** | **181** | **62.4** | 1501 | 62.3 | 3164 | 49.2 |

Bold: p<0.05.

Turning to the analysis of covert compared with overt use among recent and current contraceptive users, bivariate analysis indicated that at the individual level, covert use was associated with women's age, with covert users being younger than their counterparts who were overt users (15–24 years: 25.5% vs 19.4%, p = 0.013) and less educated (no formal/primary education: 36.6% vs 22.6%, p<0.001). Covert use was also associated with parity and religion, while at the couple level, covert use was associated with wealth (highest wealth tertile: 30.0% vs 49.9%, p<0.001) and family type (monogamous: 41.0% vs 63.0%, p<0.001).

At the community level, covert users resided in geographic clusters with higher prevalence of contraceptive use compared to non-users (middle tertile: 34.1% vs 29.7%; highest tertile: 51.4% vs 18.5%; p<0.001) (Table 2). Furthermore, compared to non-users, covert users resided in geographic clusters where greater proportions of women worked outside the home (middle tertile: 34.8% vs 30.7%; highest tertile: 48.3% vs 24.8%; p<0.001), reported media exposure, and received secondary education or higher. Compared with overt users, covert

Table 2. Community-level characteristics according to non-use, covert, and overt use of family planning in Nigeria.

| Community-Level Variables | Non-users of FP in need of contraception (n = 3733) | | Covert Users of FP (n = 290) | | Overt User of FP (n = 2410) | | Total (n = 6433) | |
|---|---|---|---|---|---|---|---|---|
| | n% | | n% | | n% | | N% | |
| **Female residents using modern contraception** | | | | | | | | |
| Lowest tertile | **1933** | **51.8** | **42** | **14.5** | 271 | 11.2 | 2246 | 34.9 |
| Middle tertile | **1109** | **29.7** | **99** | **34.1** | 891 | 37.0 | 2099 | 32.6 |
| Highest tertile | **691** | **18.5** | **149** | **51.4** | 1248 | 51.8 | 2088 | 32.5 |
| **Female residents working in the month preceding the survey** | | | | | | | | |
| Lowest tertile | **1663** | **44.6** | **49** | **16.9** | 490 | 20.3 | 2202 | 34.2 |
| Middle tertile | **1145** | **30.7** | **101** | **34.8** | 850 | 35.3 | 2096 | 32.6 |
| Highest tertile | **925** | **24.8** | **140** | **48.3** | 1070 | 44.4 | 2135 | 33.2 |
| **Female residents with at least secondary education** | | | | | | | | |
| Lowest tertile | **1778** | **47.6** | **51** | **17.6** | **355** | **14.7** | 2184 | 34.0 |
| Middle tertile | **1138** | **30.5** | **124** | **42.8** | **884** | **36.7** | 2146 | 33.4 |
| Highest tertile | **817** | **21.9** | **115** | **39.7** | **1171** | **48.6** | 2103 | 32.7 |
| **Female residents exposed to FP messages through media** | | | | | | | | |
| Lowest tertile | **1419** | **38.0** | **77** | **26.6** | 672 | 27.9 | 2168 | 33.7 |
| Middle tertile | **1152** | **30.9** | **100** | **34.5** | 894 | 37.1 | 2146 | 33.4 |
| Highest tertile | **1162** | **31.1** | **113** | **39.0** | 844 | 35.0 | 2119 | 32.9 |

Bold: p<0.05.

users lived in geographical clusters with lower levels of female education (lowest education cluster: 17.6% vs 14.7%; middle education tertile: 42.8% vs 36.7%; p = 0.016).

Mixed multilevel multivariate logistic regression showed that relative to non-use, covert use was associated with higher odds of having at least 2 births [AOR (95%CI): 1.78 (1.11–2.86), 2.56 (1.43–4.57) for women with 2–4 and ≥4 births respectively]; and being Catholic versus Protestant [1.55 (0.99–2.42)], though the latter was only of borderline significance. Women who were not living with a partner, compared with those in monogamous unions [4.60 (3.06–6.93)], those in the middle compared with the poorest wealth tertile [1.97 (1.28–3.02)], and those residing in urban areas [1.70 (1.14–2.55)] had higher odds of reporting covert use relative to non-use. At the community level, odds of covert use versus non-use were highest in geographical clusters where more women were current or recent contraceptive users and worked in the month preceding the survey (Table 3).

The mixed multilevel multivariate logistic model comparing covert to overt users showed that covert users had lower odds of secondary education or higher (secondary: 0.54, 0.37–0.78; tertiary: 0.57, 0.35–0.93) and lower odds of being wealthy (0.54, 0.31–0.96) compared to overt users. Covert users had higher odds of not living with a partner (5.01, 3.24–7.76) and residing in urban communities (1.56, 1.03–2.37) compared to overt users. At the community level, the odds of covert use were higher among women residing in communities with higher female employment outside the home (1.62, 0.96–2.73) relative to overt use. There was no significant difference in covert versus overt use according to community prevalence of contraceptive use, community levels of female education, or where exposure to media was ubiquitous (Table 4).

## Discussion/Conclusion

In this study, we measured the prevalence of covert use of contraception, and individual/couple and community level factors that influenced covert use among Nigerian women. To do so,

**Table 3. Multilevel model showing adjusted odds ratios and 95% confidence intervals of individual, couple and community-level factors associated with covert use compared to non use among Nigerian women in need of contraception.**

| Covert Use | Adjusted Odds Ratio | 95% CI | | P value |
|---|---|---|---|---|
| **Individual level variables** | | Lower limit | Upper limit | |
| **Age group (years)** | | | | |
| 15–24 | 1.00 | | | |
| 25–34 | 1.13 | 0.74 | 1.73 | 0.567 |
| 35–49 | 1.35 | 0.83 | 2.21 | 0.226 |
| **Educational level** | | | | |
| No formal/primary | 1.00 | | | |
| Secondary | 0.94 | 0.65 | 1.36 | 0.731 |
| Tertiary | 1.21 | 0.74 | 1.99 | 0.453 |
| **Number of births** | | | | |
| 0–1 births | 1.00 | | | |
| 2–4 births | 1.78 | 1.11 | 2.86 | 0.017 |
| 4+ births | 2.56 | 1.43 | 4.57 | 0.001 |
| **Religion** | | | | |
| Protestant Christian | 1.00 | | | |
| Catholic | 1.55 | 0.99 | 2.42 | 0.057 |
| Islam/Others | 0.96 | 0.67 | 1.38 | 0.819 |
| **Couple level variables** | | | | |
| **Family type** | | | | |
| Monogamous | 1.00 | | | |
| Never married/Separated/Divorced | 4.60 | 3.06 | 6.93 | <0.001 |
| Polygynous | 1.01 | 0.70 | 1.46 | 0.940 |
| **Wealth tertile** | | | | |
| Poorest | 1.00 | | | |
| Middle | 1.97 | 1.28 | 3.02 | 0.002 |
| Richest | 1.15 | 0.66 | 1.98 | 0.625 |
| **Residence** | | | | |
| Rural | 1.00 | | | |
| Urban | 1.70 | 1.14 | 2.55 | 0.009 |
| **Community Level Variables (Tertiles)** | | | | |
| **Current or recent FP use** | | | | |
| Lowest tertile | 1.00 | | | |
| Middle tertile | 2.56 | 1.51 | 4.33 | <0.001 |
| Highest tertile | 5.01 | 2.81 | 8.92 | <0.001 |
| **Secondary education or higher** | | | | |
| Lowest tertile | 1.00 | | | |
| Middle tertile | 0.97 | 0.56 | 1.68 | 0.922 |
| Highest tertile | 0.82 | 0.43 | 1.57 | 0.549 |
| **Females worked in past month** | | | | |
| Lowest tertile | 1.00 | | | |
| Middle tertile | 1.64 | 1.01 | 2.66 | 0.044 |
| Highest tertile | 1.76 | 1.07 | 2.89 | 0.026 |
| **Media exposure** | | | | |
| Lowest tertile | 1.00 | | | |
| Middle tertile | 0.76 | 0.49 | 1.17 | 0.214 |
| Highest tertile | 0.96 | 0.61 | 1.50 | 0.841 |
| _cons | 0.004 | 0.002 | 0.008 | <0.001 |

**Table 4.** Multilevel model showing adjusted odds ratios and 95% confidence intervals of community, couple and individual-level factors associated with covert use compared to overt use among Nigerian women in need of contraception.

| Covert Use | Adjusted Odds Ratio | 95% CI | | P value |
|---|---|---|---|---|
| Individual level variables | | Lower limit | Upper limit | |
| **Age group (years)** | | | | |
| 15–24 | 1.00 | | | |
| 25–34 | 0.83 | 0.54 | 1.25 | 0.359 |
| 35–49 | 1.07 | 0.65 | 1.75 | 0.795 |
| **Educational level** | | | | |
| No formal/primary | 1.00 | | | |
| Secondary | 0.54 | 0.37 | 0.78 | 0.001 |
| Tertiary | 0.57 | 0.35 | 0.93 | 0.024 |
| **Number of births** | | | | |
| 0–1 births | 1.00 | | | |
| 2–4 births | 1.50 | 0.90 | 2.52 | 0.119 |
| 4+ births | 1.58 | 0.84 | 2.98 | 0.156 |
| **Religion** | | | | |
| Protestant Christian | 1.00 | | | |
| Catholic | 1.26 | 0.82 | 1.94 | 0.284 |
| Islam/Others | 1.42 | 0.98 | 2.04 | 0.060 |
| **Couple level variables** | | | | |
| **Family type** | | | | |
| Monogamous | 1.00 | | | |
| Polygynous | 1.37 | 0.94 | 2.00 | 0.102 |
| Never married/Separated/Divorced | 5.01 | 3.24 | 7.76 | <0.001 |
| **Wealth tertile** | | | | |
| Poorest | 1.00 | | | |
| Middle | 1.32 | 0.83 | 2.10 | 0.245 |
| Richest | 0.54 | 0.31 | 0.96 | 0.036 |
| **Residence** | | | | |
| Rural | 1.00 | | | |
| Urban | 1.56 | 1.03 | 2.37 | 0.038 |
| **Community Level Variables (Tertiles)** | | | | |
| **Current or recent FP use** | | | | |
| Lowest tertile | 1.00 | | | |
| Middle tertile | 0.72 | 0.41 | 1.26 | 0.247 |
| Highest tertile | 0.89 | 0.49 | 1.61 | 0.695 |
| **Secondary education or higher** | | | | |
| Lowest tertile | 1.00 | | | |
| Middle tertile | 0.96 | 0.54 | 1.70 | 0.895 |
| Highest tertile | 0.77 | 0.40 | 1.47 | 0.493 |
| **Females worked in past month** | | | | |
| Lowest tertile | 1.00 | | | |
| Middle tertile | 1.38 | 0.83 | 2.29 | 0.213 |
| Highest tertile | 1.62 | 0.96 | 2.73 | 0.073 |
| **Media exposure** | | | | |
| Lowest tertile | 1.00 | | | |
| Middle tertile | 0.97 | 0.62 | 1.51 | 0.881 |
| Highest tertile | 1.20 | 0.76 | 1.92 | 0.435 |
| _cons | 0.045 | 0.020 | 0.103 | <0.001 |

we used representative data from Nigeria. Our study contributes to what we know about covert use in Nigeria because it used a representative sample of community-based Nigerian women who are in need of contraception to provide insight into how a woman's contraceptive use category may reflect her autonomy in achieving her reproductive goals.

Overall, our results are generally compatible with previous research on covert use. Prevalence of covert use in this study was similar to that from a study conducted in Enugu [15], but slightly lower than other Nigerian studies from Ibadan [16] and Ilorin [17], reported to be 6.8% and 7.2% respectively. These other studies were clinic-based, and reflect more localized subpopulations of family planning users within a single Nigerian state while the present study is community-based and represents the experiences of women across 8 states. Additionally, our study is more diverse, and more representative of Nigerian women in need of contraception. To further buttress this point, the present analysis suggests that covert use varies significantly by community and according to women's individual circumstances, as factors informing Nigerian women's decision to use FP covertly occurred at both the community and couple/individual level. Women who reside in communities where more women are empowered in terms of working outside the home have higher odds of reporting covert FP use, irrespective of the comparison group. Although this association was strong when comparing covert users to non-users, it was only of borderline significance when comparing covert to overt users in the final regression models. This finding may be related to external influences such as social networks in the workplace that encourage women to develop and display more self-efficacy, such that even when they are unable to discuss with their partner, or to arrive at a mutual agreement following a discussion, they can still act on their own to obtain a FP method. Additionally, women who reside in communities where more women used FP report more covert than non-use, possibly as a result of an enabling environment that normalizes use and facilitates access to FP.

Anecdotally and traditionally, the status of Nigerian women is said to be higher as they get married, grow older, and have children, and our findings suggest that women with these characteristics are more likely to use FP covertly than not at all, suggesting that covert use reflects a degree of empowerment for women in need of FP that is higher than their counterparts who need a method but do not use one. Even though our results do not reach statistical significance to show an association between older age (35–49 years) and more covert versus non-use, or covert versus overt use, or an association between high parity and more covert versus overt use, the effect estimates are in the expected positive direction. Women using contraception covertly are likely less empowered than those using contraception overtly. This latter group is generally more likely to be in union, more educated and wealthier, all of which are symbols of higher status for women in the Nigerian setting, and in keeping with findings of studies from similar settings [18,19]. These qualities may have enabled women to broach a discussion about FP use with their partners, either because of more familiarity and a sense of being equal with their partners, or because they have more understanding partners who may themselves create an enabling environment to discuss reproductive health matters, including family planning. It is also possible that the topic may be broached by either partner as a result of the economic implications of bearing and rearing additional children. Findings from clinic-based mixed-methods studies conducted in Ghana and Uganda were similar. The researchers found that being single (never married/separated/divorced), compared with being married/cohabiting was an independent predictor of reporting covert use [5,18]. Qualitative research suggests that single women may use contraception covertly because they are uncertain about the future of the relationship they are in and feel no obligation to disclose their chosen line of action [5,10].

Muslims were more likely to report covert than overt use, and this may be as a result of lack of clarity on what the Quran says [26], or what their husbands want, as they may not have the

courage to discuss FP, especially given the pronatalist expectation of others in their faith, and the possible wife rivalry linked to polygyny. Although data from PMA2020 do not address social expectations, the present analysis suggests that women who may not be socially expected to use modern contraception, such as Catholics and Muslims report more covert use than Protestant Christians. Catholics are encouraged to use natural FP methods only, and our data show that they did so more than followers of other religions, but many in this study also reported use of modern methods, especially emergency contraception, oral pills, and injectables. When used overtly, Catholics were more likely to report use of male condoms, withdrawal and implants (data not shown). These results reinforce the need for confidential services to meet women's family planning needs, including within faith-based organizations.

Women who reside in urban areas report more covert use than those who live in rural areas, contrary to what was reported by some other older research from Uganda [22]. This finding may be contextual and may be more likely in countries with more family planning use, unlike Nigeria where use remains relatively low. However, studies that looked at use overall without disaggregating into covert and overt use generally report more use in urban than rural communities [27,28], even though this may not reach statistical significance at the multivariable level [29].

Although some researchers report that the pill is harder to conceal due to daily use, and thus not a good candidate method for covert use [19], women who used a method covertly in the present study reported the pill more frequently than any other method. As described by women in the qualitative study by Kibira and colleagues (2020), covert use in the present study was facilitated by the use of female-controlled methods that are easy to conceal, such as injectables, implants, and emergency contraception [10]. Our study also aligns with that of Wolff and colleagues in Uganda who reported from their mixed-methods study that women used periodic abstinence, pills and traditional methods covertly and these were methods that women normally used without informing their partners or seeking their approval [6]. Additionally, qualitative research suggests that women who do not live with their partners report more covert use of traditional methods like fertility awareness approaches, but this is harder to conceal for women in union who fear discovery [10]. Baiden and colleagues found from qualitative research in Ghana that women go to great lengths to conceal family planning use, including dropping their clinic card in their mother's house before heading home [5]. The women in this Ghanaian study suggested that health workers should support women to attain their reproductive goals, even if this had to be done through covert use of family planning methods [5].

Although this study has many strengths, these results must be interpreted in the context of a few limitations. First, this paper only presents the reports of women regarding covert use, but the existing body of research suggests that this mode of direct measurement may only tell part of the story, as discordant reports are not uncommon when the perspective of the male partner is sought. For instance, Wolff and colleagues found from focus group discussions that men who know about their partner's menstrual cycle may periodically abstain from having sex with her to avoid a pregnancy because he thinks she would not be interested in birth control options [6]. Choiryyah and Becker (2018) argue that neither direct nor indirect measurements of covert use are accurate as the former is an underestimate while the latter produces an overestimate [14]. Second, the fact that the factors that are associated with covert use occur at multiple levels suggests there is some nuance that this study was unable to account for. For instance, qualitative research from both women and men residing in Ethiopia, Nigeria, and Uganda suggests that women may be unable to assert their reproductive preferences in a relationship that is failing as a result of gender power imbalance, intimate partner violence, and relationship instability as a result of suspected infidelity, or no financial support from the partner [10,13], factors that the present study does not address. However, a single study cannot tell

the whole story, and the concordance of our findings with the existing body of literature provides reassurance that our inability to account for such nuance does not detract from the importance of our findings.

Despite these limitations, this study adds to our current understanding of women's contraceptive use, identifying factors associated with covert use versus non-use and covert use versus overt use, which not only reside in individual women, but are also related to an enabling environment, where women have more economic and educational opportunities, and exercise greater control over their fertility. These results, based on a representative sample of the female population in Nigeria, have implications that can be of use to researchers, program managers, non-governmental organizations, and the government. For women in need of contraception, more covert use than non-use in communities where more women work outside the home and where more women use FP reflects some degree of empowerment, thus increasing awareness of FP through workplace programs and peer sharing may make it possible for women who need FP but are unable or unwilling to discuss this with their partners to obtain the information and services they need. Anecdotally, Nigerian women do not usually go to FP clinics with their partners, so programs and NGOs offering FP services need to train and retrain FP providers to tailor messages appropriately for women who live with their partners, and those who do not, in order to optimize their ability to make informed decisions. Further research on the contextual differences in the perception and practice of covert use may increase our understanding and help to provide more informed counseling messages for women who seek FP. More research using qualitative methods, and a life history approach may also provide a more in-depth understanding of covert use of FP among women in a given context.

Overall, covert use was reported by 4.5%, while overt use was reported in 37.5% of Nigerian women in this study in need of a family planning method. However, many (58.0%) Nigerian women in need of FP remain non-users. Our data suggest that covert use represents some degree of empowerment among women in need of FP, but who are unable to use more overtly due to barriers. Overt use represents the ability to overcome such barriers. Having at least 2 children, being Catholic, not living with her male partner, being in the middle wealth tertile, and residing in an urban area were all associated with more covert use, while there were lower odds of covert compared with overt use among women with at least secondary education and those who were in the richest wealth tertile. At the community level, covert use was more common where more women used FP relative to non-use and where more women worked outside the home. Programs should ensure that their providers and facilities are supportive of all women who need FP, whether this need is satisfied with covert or overt use.

## Author Contributions

**Conceptualization:** Funmilola M. OlaOlorun, Caroline Moreau.

**Formal analysis:** Funmilola M. OlaOlorun.

**Methodology:** Funmilola M. OlaOlorun, Caroline Moreau.

**Visualization:** Funmilola M. OlaOlorun.

**Writing – original draft:** Funmilola M. OlaOlorun.

**Writing – review & editing:** Funmilola M. OlaOlorun, Philip Anglewicz, Caroline Moreau.

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
