## [Decision Letter · Decision Letter 0]

13 Sep 2020

PONE-D-20-19653

From non-use to covert and overt use of contraception: identifying community and individual factors informing Nigerian women’s degree of contraceptive empowerment

PLOS ONE

Dear Dr. OlaOlorun,

Thank you for submitting your manuscript to PLOS ONE. After careful consideration, we feel that it has merit but does not fully meet PLOS ONE’s publication criteria as it currently stands. Therefore, we invite you to submit a revised version of the manuscript that addresses the points raised during the review process.

We look forward to receiving your revised manuscript.

Kind regards,

Catherine S. Todd

Academic Editor

PLOS ONE

Additional Editor Comments:

This is a well-written and interesting paper and I agree with the reviewers. If the authors can make the suggested edits and clarifications, this manuscript should be acceptable for publication.

Journal Requirements:

2. Please note that in order to use the direct billing option the corresponding author must be affiliated with the chosen institute. Please either amend your manuscript or remove this option (via Edit Submission).

Reviewers' comments:

Reviewer's Responses to Questions

**Comments to the Author**

1. Is the manuscript technically sound, and do the data support the conclusions?

Reviewer #1: Yes

Reviewer #2: Yes

2. Has the statistical analysis been performed appropriately and rigorously? 

Reviewer #1: Yes

Reviewer #2: Yes

3. Have the authors made all data underlying the findings in their manuscript fully available?

Reviewer #1: Yes

Reviewer #2: Yes

4. Is the manuscript presented in an intelligible fashion and written in standard English?

Reviewer #1: Yes

Reviewer #2: Yes

5. Review Comments to the Author

Reviewer #1: This study was well formulate with a clear and important research question. The methodology appeared sound with appropriate statistical analysis. The relevant data was presented in a clear and understandable manner. The authors set out to determine the prevalence of covert use and factor that influence covert use.

Overall, I found that the authors answered the first part of the question of determining the prevalence, but the second part was less well answered. The factors identified seemed more to be associated with covert use than influences on covert use. This could be a restriction in the original survey data and covert related categories than study design by the authors.

Some interesting and unusual findings were presented for covert use, such as the preference for the pill or the high covert use in setting where other women are using contraception overtly.

I only have a few comments/suggestions that the authors may consider to improve the strength of this paper:

1. My primary comment is about the use of the word empowerment in relation to covert use. Various results presented in the study compare non-use with covert use - this can show a somewhat skewed perspective on the empowerment of covert use. Covert use remains a reflection of disempowerment of women, in any society. I feel that this could be phrased more appropriately and the dangerous of covert use should be explicitly stated. Covert use is a threat to continued use of contraception, and therefore is a risk factor for unintended pregnancies. While it is true that covert users may be more empowered than non-users, this should be phrased with caution.

2. The conclusion in line 315-318 is not well supported by the data. While it is clear that male partners will not be informed of the covert use, it is not clear that the overt users in the community are aware of covert users using contraception. Covert users may well feel inferior or scared to reveal their contraceptive use to overt users, if the overt users occupy a more powerful position in society (as the authors claim). These covert users tended to be single compared to overt users who are married or in relationships. This power dynamic can have a significant influence on younger, single women overtly using contraception.

2. Please check the titles of the tables - some where a little misleading, especially table 3 and 4 where the comparative results were not included. There are also four columns of data in each table that are not clearly labelled.

3. Lines 137-140 will benefit from a reference.

Overall, I think that this is a well written paper on a very important topic that requires further exploration.

Reviewer #2: This manuscript is a very interesting and well-written exploration of multi-level determinants of covert use among a representative sample of women in Nigeria. This study is very timeline given the persistent low rates of contraceptive use in Nigeria. Additionally, given increased focus on male involvement in family planning programs, it is important to understand correlates of women’s decisions to use contraception covertly in order to promote related practices in a safe and women-centered way. My few comments are as follows:

1. It could be useful to provide a brief definition of covert use at the start of the background prior to describing why women may have chosen to use contraception in this way.

2. In the background, it is briefly mentioned that reproductive coercion, partner violence, and misperception of partners’ attitudes are reasons for covert use. If available, it may be useful for the authors to provide some information on the reasons women choose to use covertly in this specific cultural context, as this may vary depending on the setting.

3. In lines 184 – 187, the way authors have defined whether participants had need for contraception could be clarified. As it is written, it is currently unclear whether those that had not recently used contraception and were sexually active were considered to have need regardless of desire to become pregnant or only if they were not trying to become pregnant.

4. The authors mention in the discussion that age and parity are factors that generally grant women status and as a results, increases their empowerment. It may be interesting to comment on why in this study age was not found to relate to covert use vs no use or covert use vs overt use and why parity was not found to associate with covert vs overt use.

5. While this study certainly lends important detail to the literature on covert use, because fertility goals and contraceptive use preferences/intentions may be dynamic it may be important to take care not to refer to nonuse, covert use, and overt use as a continuum as in lines 397-401. This may be particularly true given that data used were cross-sectional.

6. PLOS authors have the option to publish the peer review history of their article (what does this mean?). If published, this will include your full peer review and any attached files.

Reviewer #1: **Yes: **Yolandie Kriel

Reviewer #2: No

---

## [Author Response · Author response to Decision Letter 0]

4 Oct 2020

PONE-D-20-19653R1

 From non-use to covert and overt use of contraception: identifying community and individual factors informing Nigerian women’s degree of contraceptive empowerment

Responses to Reviewers’ Comments

Reviewer #1 Comments

1.

“My primary comment is about the use of the word empowerment in relation to covert use. Various results presented in the study compare non-use with covert use - this can show a somewhat skewed perspective on the empowerment of covert use. Covert use remains a reflection of disempowerment of women, in any society. I feel that this could be phrased more appropriately and the dangerous of covert use should be explicitly stated. Covert use is a threat to continued use of contraception, and therefore is a risk factor for unintended pregnancies. While it is true that covert users may be more empowered than non-users, this should be phrased with caution.” 

We appreciate this perspective and have added information on the threats associated with covert use in lines 150-155 to address this important nuance.

2.

“The conclusion in line 315-318 is not well supported by the data. While it is clear that male partners will not be informed of the covert use, it is not clear that the overt users in the community are aware of covert users using contraception. Covert users may well feel inferior or scared to reveal their contraceptive use to overt users, if the overt users occupy a more powerful position in society (as the authors claim). These covert users tended to be single compared to overt users who are married or in relationships. This power dynamic can have a significant influence on younger, single women overtly using contraception.” 

Thanks for sharing this perspective. We have edited the statement to read, “Additionally, women who reside in communities where more women used FP report more covert than non-use, possibly as a result of an enabling environment that normalizes use and facilitates access to FP.” Please see lines 331-333.

3.

“Please check the titles of the tables - some where a little misleading, especially table 3 and 4 where the comparative results were not included. There are also four columns of data in each table that are not clearly labelled.” 

Thank you for this observation. Table and column titles have been edited accordingly.

4.

“Lines 137-140 will benefit from a reference.”

A reference has been inserted. The updated lines are 151-155.

\f

Reviewer #2 Comments

1.

“It could be useful to provide a brief definition of covert use at the start of the background prior to describing why women may have chosen to use contraception in this way.”

A definition has been added on lines 71-72: “Covert use, the use of a contraceptive method without the knowledge of a woman’s partner, may represent…”

2.

“In the background, it is briefly mentioned that reproductive coercion, partner violence, and misperception of partners’ attitudes are reasons for covert use. If available, it may be useful for the authors to provide some information on the reasons women choose to use covertly in this specific cultural context, as this may vary depending on the setting.” 

Motivations for covert use in this specific context have been added, on lines 110-114.

3.

“In lines 184 – 187, the way authors have defined whether participants had need for contraception could be clarified. As it is written, it is currently unclear whether those that had not recently used contraception and were sexually active were considered to have need regardless of desire to become pregnant or only if they were not trying to become pregnant.” 

A statement on lines 202 to 204 has been added to clarify: “Women who desired a child within the 2-year period following the survey were not considered to be in need for contraception and were excluded from the analytic sample.”

4.

“The authors mention in the discussion that age and parity are factors that generally grant women status and as a results, increases their empowerment. It may be interesting to comment on why in this study age was not found to relate to covert use vs no use or covert use vs overt use and why parity was not found to associate with covert vs overt use.”

A comment has been added in lines 339-342: “Even though our results do not reach statistical significance to show an association between older age (35-49 years) and more covert versus non-use, or covert versus overt use, or an association between high parity and more covert versus overt use, the effect estimates are in the expected positive direction.”

5.

“While this study certainly lends important detail to the literature on covert use, because fertility goals and contraceptive use preferences/intentions may be dynamic it may be important to take care not to refer to nonuse, covert use, and overt use as a continuum as in lines 397-401. This may be particularly true given that data used were cross-sectional.”

We appreciate this perspective and have changed the word “continuum” to “spectrum” throughout the paper. We agree that the data we have may not be able to truly reflect a continuum, given its cross sectional nature.

---

## [Decision Letter · Decision Letter 1]

2 Nov 2020

From non-use to covert and overt use of contraception: identifying community and individual factors informing Nigerian women’s degree of contraceptive empowerment

PONE-D-20-19653R1

Dear Dr. OlaOlorun,

We’re pleased to inform you that your manuscript has been judged scientifically suitable for publication and will be formally accepted for publication once it meets all outstanding technical requirements.

Kind regards,

Catherine S. Todd

Academic Editor

PLOS ONE

Additional Editor Comments (optional):

I thank the authors for their revisions and improvements of this interesting article.

Reviewers' comments:

Reviewer's Responses to Questions

**Comments to the Author**

1. If the authors have adequately addressed your comments raised in a previous round of review and you feel that this manuscript is now acceptable for publication, you may indicate that here to bypass the “Comments to the Author” section, enter your conflict of interest statement in the “Confidential to Editor” section, and submit your "Accept" recommendation.

Reviewer #1: All comments have been addressed

Reviewer #2: All comments have been addressed

2. Is the manuscript technically sound, and do the data support the conclusions?

Reviewer #1: Yes

Reviewer #2: (No Response)

3. Has the statistical analysis been performed appropriately and rigorously? 

Reviewer #1: Yes

Reviewer #2: (No Response)

4. Have the authors made all data underlying the findings in their manuscript fully available?

Reviewer #1: Yes

Reviewer #2: (No Response)

5. Is the manuscript presented in an intelligible fashion and written in standard English?

Reviewer #1: Yes

Reviewer #2: (No Response)

6. Review Comments to the Author

Reviewer #1: The authors addressed my previous comments satisfactorily. I think this is a informative article that sheds light onto the complexity of using contraception.

Reviewer #2: (No Response)

7. PLOS authors have the option to publish the peer review history of their article (what does this mean?). If published, this will include your full peer review and any attached files.

Reviewer #1: **Yes: **Yolandie Kriel

Reviewer #2: No

---

## [Editor Report · Acceptance letter]

10 Nov 2020

PONE-D-20-19653R1 

From non-use to covert and overt use of contraception: identifying community and individual factors informing Nigerian women’s degree of contraceptive empowerment 

Dear Dr. OlaOlorun:

I'm pleased to inform you that your manuscript has been deemed suitable for publication in PLOS ONE. Congratulations! Your manuscript is now with our production department. 

Kind regards, 

on behalf of

Dr. Catherine S. Todd 

Academic Editor

PLOS ONE